# Simultaneous Determination of N^ε^-(carboxymethyl) Lysine and N^ε^-(carboxyethyl) Lysine in Different Sections of Antler Velvet after Various Processing Methods by UPLC-MS/MS

**DOI:** 10.3390/molecules23123316

**Published:** 2018-12-14

**Authors:** Rui-ze Gong, Yan-hua Wang, Yu-fang Wang, Bao Chen, Kun Gao, Yin-shi Sun

**Affiliations:** 1Institute of Special Animal and Plant Sciences, Chinese Academy of Agricultural Sciences, Changchun 130112, China; 82101172456@caas.cn (R.-z.G.); yhwangsdlc@126.com (Y.-h.W.); wangyufang_jl@163.com (Y.-f.W.); 15543598331@163.com (B.C.); 13356954028@163.com (K.G.); 2College of Chinese Material Medicine, Jilin Agricultural University, Changchun 130112, China

**Keywords:** advanced glycation end-products (AGEs), N^ε^-(carboxymethyl) lysine (CML), N^ε^-(carboxyethyl) lysine (CEL), antler velvet processing, UPLC-MS/MS

## Abstract

N^ε^-(Carboxymethyl) lysine (CML) and N^ε^-(carboxyethyl) advanced glycation end-products (AGEs) and are frequently used as markers of AGE formation. AGEs, such as CML and CEL, have harmful effects in the human body and have been closely linked to many diseases such as diabetes and uremia. However, details on the contents of CML and CEL after applying different antler velvet processing methods are lacking. In this research, a robust lysine (CEL) are two typical UPLC-MS/MS method has been developed for the simultaneous determination of CML and CEL in various sections of antler velvet processed with different methods. In addition, factors affecting the CML and CEL contents are discussed. The CML contents of antler velvet after freeze-drying, boiling, processing without blood, and processing with blood were 74.55–458.59, 119.44–570.69, 75.36–234.92, and 117.11–456.01 μg/g protein, respectively; the CEL contents were 0.74–12.66, 11.33–35.93, 0.00–6.75, and 0.00–23.41 μg/g protein, respectively. The different contents of CML and CEL in the different samples of antler velvet result from the different interactions of the protein and lysine at different temperatures. These data can be used to estimate the potential consumer intake of CML and CEL from antler velvet and for guiding producers on how to reduce the production of CML and CEL.

## 1. Introduction

Antler velvet is a representative animal medicinal material and dietary supplement that has been an important part of traditional Chinese medicine for thousands of years in China, Korea, and Southeast Asian countries [1,2,3]. It has various pharmacological effects, such as anti-oxidation and anti-osteoporosis properties [4,5,6]. Fresh antler velvet is rich in nutrients, such as proteins and amino acids; these are highly susceptible to spoilage if the antler velvet is not processed promptly. Based on the methods of processing and consumption, antler velvet can be classified as processed with blood or without blood, as boiled or freeze-dried, and as wax slices, powder slices, gauze slices, or bone slices. Different processing methods and sections of the antler velvet have different influences on the bioactive components and pharmacological activities [4,5,6]; therefore, the processing conditions are crucial for antler velvet’s dietary and medical functions.

During the processing and storage of antler velvet, amino compounds (e.g., proteins, amino acids) react with carbonyl compounds (e.g., reducing sugars, lipid oxidation products) to randomly form advanced glycation end-products (AGEs) by the Maillard reaction [7]. AGEs have been shown to be detrimental to human health, being closely linked to many conditions such as diabetes, Alzheimer’s disease, atherosclerosis, renal diseases, and aging [8,9,10,11,12]; with the accumulation of AGEs in the human body, the probability of people suffering the above-mentioned chronic degenerative diseases will increase greatly. Food is the main source of AGEs, especially high-fat and high-sugar foods [8,11,13], although human bodies can also produce some AGEs by themselves.

N^ε^-(Carboxymethyl) lysine (CML) and N^ε^-(carboxyethyl) lysine (CEL) are two typical AGEs and are frequently used as markers of AGE formation in foods [14,15,16]. However, the determination of CML and CEL in antler velvet has not been reported. The contents of CML and CEL in antler velvets are affected by the matrix and processing conditions [17,18]. Antler velvet rich in amino compounds and carbonyl compounds may contribute more CML and CEL than other foods [19,20]. Therefore, information on the CML and CEL contents in processed antler velvets is essential to estimate the potential consumer intake of AGEs from antler velvet.

Since CML and CEL have no UV absorption and fluorescence properties, enzyme-linked immunosorbent assay (ELISA), high-performance liquid chromatography (HPLC), gas chromatography–mass spectrometry (GC-MS), and high-performance liquid chromatography-mass spectrometry (HPLC-MS) techniques have previously been used to determine their contents [21,22,23,24,25,26,27,28]. The ELISA method requires specific antibodies, and the sensitivity is greatly affected by the matrix effect of the sample, which can cause large errors [21,22,23]. HPLC and GC-MS typically require pre-column derivatization, which is cumbersome and reduces sensitivity [24,25,26]. HPLC-MS has the advantages of simple operation, repeatability, and stability; therefore, HPLC-MS has usually been used to determine CML and CEL contents [14,16,27].

In this research, we have developed a robust UPLC-MS/MS method to simultaneously determine CML and CEL contents in various sections of antler velvet processed with different methods. The CML and CEL contents in the various samples were determined by a validated method, which may contribute to the assessment of AGEs in antler velvets. This study provides a foundation and valuable reference for safe antler velvet processing and provides a basis for the development of recommended antler velvet dosages.

## 2. Results and Discussion

### 2.1. Sample Pretreatment

Preparation of the antler velvet samples consisted of processing, segmenting, grinding, defatting, reduction, hydrolysis, and SPE. According to the processing method and consumption, antler velvet samples were classified as boiled or freeze-dried and processed with or without blood; samples were then divided into wax, powder, gauze, and bone slices. Because CML and CEL can be generated via peroxidation of the antler velvet lipid content, it was important to remove lipids from the antler velvet samples to prevent overestimation in the results. Before hydrolysis, 0.1 N sodium borohydride was applied for 12 h to reduce the amadori (e.g., fructose–lysine) and lipid oxidation products, thus preventing the formation of CML and CEL during acid hydrolysis [28]. The samples were subjected to SPE by using a C_18_ Sep-Pak cartridge (Sepax technology, Cork, Ireland; 500 mg, 6 mL), to remove impurities from the sample.

### 2.2. Optimization of Chromatography Conditions

CML and CEL are highly polar compounds and are difficult to retain in most reversed-phase columns. Researchers have usually analyzed these substances with a C18 column by using nonafluoropentanoic acid (NFPA) as the eluent. However, NFPA can lead to a low-pH (~2) mobile phase, which may result in deterioration of the reversed-phase column [29]. To avoid the use of NFPA, we developed a UPLC-MS/MS method to separate CML and CEL with a WATERS CORTECS HILIC UPLC column. HILIC uses the separation principle of affinity chromatography to maximize the retention and separation of highly polar compounds, relative to other columns. The elution effects of methanol and acetonitrile were assessed: acetonitrile/water (30:70 *v/v*) and methanol/water (30:70 *v/v*) mixtures were used as mobile phases, and the flow rate was 0.3 mL/min. In comparison to chromatograms with the methanol mobile phase, UPLC-MS chromatograms using water and acetonitrile as the eluent have better spectrum peak symmetries and fewer miscellaneous peaks.

The UPLC-MS chromatograms of standard solutions (a) and antler velvet samples (b) are shown in Figure 1. The retention times of CML and CEL in the antler velvet samples were consistent with those of the standard solutions, and no peak interference was observed. The column was very stable and robust without obvious shifts in the retention times throughout the experimental procedure.

### 2.3. Method Validation

The developed method was validated by assessing the CML and CEL contents in antler velvet samples and considering the selectivity, linearity, precision, and accuracy. The fragmentation pattern of CML and CEL indicated two major product ions at *m*/*z* 84 and 130, with the most intense peak at *m*/*z* 130. The two sample ions were used for quantitation in MRM mode.

As shown in Table 1, the correlation coefficients (R^2^) were both greater than 0.99. The linear range (20–3500 ng/mL) was sufficiently wide to assess the CML and CEL contents in the present antler velvet samples. The limit of detection (LOD) and limit of quantitation (LOQ) were defined as the concentrations (ng/g) at which the signal-to-noise ratios of the peaks of interest were 3 and 10, respectively (Table 1).

The antler velvet samples were extracted in triplicate and analyzed by using the developed UPLC-MS/MS method. The relative standard deviations of intra-day precision for CML and CEL were 3.32% and 3.08%, respectively, and those of inter-day precision were 3.14% and 3.53%, respectively. The coefficients of variation obtained for the reproducibility tests described above were less than 5%. The recoveries of exogenous CML and CEL added to antler velvet samples were determined at three concentrations (low, intermediate, and high): 30, 300, and 3000 ng/mL, respectively. Recovery experiments were conducted five times for each concentration, affording values for CML and CEL of 93.22–97.42% and 91.84–95.43%, respectively.

### 2.4. CML and CEL Contents in Processed Antler Velvet

#### 2.4.1. CML Contents in Different Sections of Antler Velvet with Different Processing Methods

The CML contents in different sections of antler velvet processed with different methods are shown in Table 2. The CML contents in freeze-dried and boiled antler velvet were 74.55–458.59 and 119.44–570.69 μg/g protein, respectively. The CML contents in antler velvet processed without blood and with blood were 75.36–234.92 and 117.11–456.01 μg/g protein, respectively. These results indicate that antler velvet protein is glycosylated, to a considerable extent, relative to other processed foods such as fried chicken breast (12.34–90.52 μg/g protein) and processed meat and fish (44.53–167.60 μg/g protein) [15,16,30].

The CML contents of freeze-dried antler velvet were significantly lower than those of the corresponding sections of boiled antler velvet (*P* < 0.01). This suggests that temperature can affect the formation of CML; specifically, high-temperature processing can produce more CML. The high content of CML in freeze-dried antler velvet is endogenous. The CML contents of antler velvet processed without blood were significantly lower than those of the corresponding sections of antler velvet processed with blood (*P* < 0.01). The reason may be that the antler velvet processed without blood was subjected to physical centrifugal discharge of the blood. Blood contains many reducing sugars, amino acids, and proteins that can react to generate CML during processing.

Wax pieces had the highest CML content, and bone pieces had the lowest. The differentiation capacity was faster closer to the top of the antler velvet; therefore, higher contents of reducing sugars, proteins, and amino acids may exist in the antler velvet to produce CML. Closer to the bottom of the antler velvet, the degree of ossification is higher; therefore, the reducing sugar, protein, and amino acid contents are lower and produce less CML.

In summary, by comparing the CML contents from different sections of antler velvet after different processing methods, the CML contents of freeze-dried antler velvet and antler velvet processed without blood were found to be lower than those of the corresponding areas of boiled antler velvet and antler velvet processed with blood; wax pieces were more likely to produce CML than other sections.

#### 2.4.2. CEL Contents in Different Sections of Antler Velvet with Different Processing Methods

The CEL contents in different sections of antler velvet processed with different methods are shown in Table 3. The CEL contents in freeze-dried and boiled antler velvet were 0.74–12.66 and 11.33–35.93 μg/g protein, respectively. The CEL contents in antler velvet processed without blood and with blood were 0–6.75 and 0–23.41 μg/g protein, respectively. The results indicate that the CEL contents in antler velvet are similar to those in other processed foods such as bread [27].

In general, the variation tendency for CEL in different sections of antler velvet after different processing methods was similar to that for CML, despite the CEL contents of antler velvet being lower than the CML contents or undetected. The CEL contents in freeze-dried antler velvet were significantly lower than those in the corresponding sections of boiled antler velvet (*P* < 0.01); the CEL contents of antler velvet processed without blood were significantly lower than those of the corresponding sections of antler velvet processed with blood (*P* < 0.01). Wax pieces had the highest content of CEL, and bone pieces had the lowest. As described above for CML, the differences in temperature and contents of proteins, amino acids, and reducing sugars are the main reasons for the different CEL contents after different processing methods.

### 2.5. Protein and Lysine Contents in Processed Antler Velvet

The CML and CEL in antler velvet exist in the combined state and the free form, among which the combined state is the most common. The CML and CEL contents and protein contents of samples are closely related in the combined state [23]. Therefore, the CML and CEL contents can be expressed in units of protein. The Dumas combustion method was used to determine the protein contents in different sections of antler velvet after different types of processing; the results are shown in Table 4.

It was found that there was no significant difference (*P* > 0.05) in protein content between the same sections of freeze-dried and boiled antler velvet. Additionally, sections of antler velvet processed with blood had significantly higher protein contents than those processed without blood (*P* < 0.05), because the blood, which contains protein, had been retained during the processing.

The protein contents were different among the different sections. Wax pieces had significantly higher contents of protein than the other sections (*P* < 0.01). The reason may be that wax pieces from the antler tip had more meristem tissue, which promotes the expression of protein [20]. Protein is the Maillard reaction substrate, and its distribution in different sections of antler velvet subjected to different processing methods is a leading factor for the differences in the CML and CEL contents, consistent with the results in Section 2.4.

An automatic amino acid analyzer was used to determine the lysine contents in different sections of antler velvet processed differently; the results are shown in Table 5.

No significant differences were observed in the lysine contents between the same sections of antler velvet processed with different methods (*P* > 0.05). The lysine contents in different sections of antler velvet were different. Wax slices had significantly higher lysine contents than the other sections (*P* < 0.01), and there was no significant difference among powder, gauze, and bone slices (*P* > 0.05). CML and CEL are two lysine derivatives; the contents of lysine can reflect the degree of reaction of different samples. The results are roughly consistent with the results for CML and CEL contents discussed in Section 2.4.

### 2.6. Factors Influencing CML and CEL Contents in Differently Processed Antler Velvets

The differences in CML and CEL contents in different sections of antler velvet with different processing methods are caused by different degrees of the Maillard reaction. Factors that affect the Maillard reaction include the processing temperature and the contents of reducing sugars, unsaturated fatty acids, amino acids, and protein [17,18,31]. In addition, vitamins and inorganic ions in food can inhibit or promote the Maillard reaction [32].

The study determined the contents of CML and CEL in different sections of antler velvet with different processing methods and found that antler velvet boiled at high temperature produced more CML and CEL than that freeze-dried at low temperature: high temperatures exacerbate the Maillard reaction and cause boiled antler velvet to produce more CML and CEL. The lysine and protein contents were different in different sections of antler velvet after different processing methods. Lysine and proteins are substrates for the Maillard reaction; therefore, their concentration determines the extent of the Maillard reaction. Relative to antler velvet processed without blood, antler velvet processed with blood, which is rich in lysine and protein, may produce more CML and CEL. Similarly, wax pieces rich in lysine and protein are more likely to produce CML and CEL than other sections.

According to the literature [32], the contents of vitamins and inorganic ions in food, such as vitamin B, vitamin C, and calcium, magnesium, and ferric ions, can affect the Maillard reaction and inhibit or promote the production of AGEs. Among these factors, vitamin B, vitamin C, calcium ions, and magnesium ions can inhibit the Maillard reaction, especially magnesium ions; ferric ions can promote the occurrence of this reaction. Therefore, the contents of vitamins and inorganic ions in different sections of antler velvet after different processing methods can also affect the CML and CEL contents.

To summarize, the differences in CML and CEL contents in different sections of antler velvet after different processing methods are the result of the combined action of lysine, proteins, vitamins, and inorganic ions at different temperatures.

## 3. Materials and Methods

### 3.1. Materials

CML, CEL, and trifluoroacetic acid (TFA) were purchased from Sigma-Aldrich (San Francisco, CA, USA). The purities of these standards were above 99%. Lysine, ninhydrin (NIN), and a citric acid buffer solution were purchased from Hitachi Inc. (Hitachi Co., Osaka, Japan). Acetonitrile, HPLC-grade, was purchased from Fisher-Scientific (Waltham, MA, USA). C_18_ Sep-Pak^®^ SPE tubes were purchased from Sepax (Sepax technology, Cork, Ireland). Ultrapure water was obtained by using a super-pure water system (Water Purifier Co. Ltd., Chengdu, China). All other reagents were of analytical grade and were purchased from Sinopharm Chemical Reagent Co. Ltd (Beijing, China).

### 3.2. Sources and Preparation of Antler Velvet

Antler velvet (*Cervi Cornu Pantotrichum*) was collected in Shuangyang, Jilin Province, China, and identified by Dr. C.Y. Li from the Chinese Academy of Agricultural Sciences Institute of Special Animal and Plant Sciences.

### 3.3. Preparation of Processed Antler Velvet

In accord with the classification of commercially available antler velvet, boiling, freeze-drying, processing with blood, and processing without blood were chosen for this study. Six pairs of antler velvet samples were randomly selected and processed with blood or without blood for comparison, and another six pairs were randomly selected and processed by boiling or freeze-drying for comparison. Antler velvet was boiled for 1 min in boiled water, followed by high-temperature (75 °C) baking for multiple 2 h cycles until dry. During the freeze-drying process, the antler velvet was directly frozen to dryness. For the boiling process without blood, the antler velvet was prepared by removing the blood by centrifugation, whereas no blood removal was performed for the samples processed with blood.

### 3.4. Preparation of Antler Velvet Slices

Three pairs of antler velvet samples with blood and without blood were randomly selected for analysis, and three pairs of boiled and freeze-dried antler velvet were crushed whole. The remaining six pairs of antler velvet were divided into wax slices, powder slices, gauze slices, and bone slices based on morphological and microscopic characteristics (Figure 2) [20]; these samples were segmented, sliced, crushed, sieved, bagged, and labeled.

### 3.5. Protein Content Analysis

The protein contents in antler velvet samples prepared by different processing methods and from different sections were determined on a Dumas nitrogen analyzer (Velp NDA 701-Monza, Brianza, Italy), according to a previous method with minor modifications [33]. The total nitrogen content was converted into the protein content by using a conversion factor of 6.25. The operating conditions of the NDA instrument were: O_2_ gas at 400 mL/min, He gas at 195 mL/min, combustion reactor at 1030 °C, reduction reactor at 650 °C, and pressure at 88.1 kPa.

### 3.6. Lysine Content Analysis

An amino acid analyzer (L-8900 System; Hitachi Co., Osaka, Japan) equipped with a visible detector was used for amino acid analysis. Analytical 2622# (4.6 mm × 60 mm) and guard 2650# (4.6 mm × 40 mm) columns were used for amino acid determination. Immediately after sample injection into the columns, an auto-sampler was used for inline derivatization by NIN post-column derivatization. The NIN-derivatized lysine were detected at 570 nm. Lysine standards (Hitachi Co., Osaka, Japan) were used for identification and quantification (external standard method). Lysine content was expressed as g/100 g of antler velvet for the different processing methods and different sections of the antler velvet.

### 3.7. Preparation of Samples

The method of reference was slightly modified in this work [16,27]. Samples of 30 mg of antler velvet (equivalent to 20 mg of protein) were defatted twice by using n-hexane (5 mL) before being reduced for 12 h at 4 °C in 0.5 M sodium borate buffer (pH 9.2, 1 mL) and 2 M sodium borohydride (0.1 M sodium hydroxide, 0.5 mL). The proteins were isolated by using a chloroform:methanol mixture (2:1 *v/v*, 1 mL), and the precipitates were mixed with 15 mL of 6 M hydrochloric acid and incubated at 110 °C for 24 h. The diluted acid hydrolysate (equivalent to approximately 600 μg of protein) was dried with a gas-blowing concentrator (Hengao, Tianjin, China) at 70 °C. The dried hydrolysate was dissolved in 1 mL ultra-pure water and then solid-phase extracted by using a C_18_ Sep-Pak^®^ (Sepax Technology, Cork, Ireland) cartridge (500 mg, 6 mL). The solid-phase extraction (SPE) column was pretreated with 3 mL of methanol and 3 mL of 0.1 M TFA at a flow rate of 1 mL/min. The sample was loaded into the pretreated SPE column at a flow rate of 0.5 mL/min and washed with 6 mL of 0.1 M TFA. Finally, the sample was eluted with 3 mL of methanol at a flow rate of 0.5 mL/min. The eluate was dried by freezing, re-dissolved in 1 mL of ultra-pure water, filtered through a 0.22-μm membrane, and stored at −20 °C prior to UPLC-MS/MS analysis.

### 3.8. UPLC-MS/MS Analysis

CML and CEL concentrations in the hydrolysates were determined by UPLC-MS/MS. Protein hydrolysates (2 μg protein, 3 μL) were injected into a WATERS CORTECS HILIC UPLC column (2.1 mm × 50 mm, 1.6 μm; Waters, Cork, Ireland) housed in a column oven at 40 °C and operated in gradient-elution mode. Solvent A was water and solvent B was acetonitrile. Gradient elution was started at 100% solvent B for 1 min; this was followed by a linear gradient from 100% to 60% solvent B in 1.5 min, holding at 60% solvent B for 1.5 min, and then returning to 100% solvent B in 2 min. The analysis was performed by using a Waters Acquity UPLC instrument (Waters, Manchester, UK) coupled to a triple quadrupole MS operating in multiple reaction monitoring (MRM) mode. The flow rate was 0.3 mL/min. The MS instrument was operated in electrospray ionization positive mode. The optimized MRM conditions are shown in Table 6. CML and CEL were quantified by using standards and by reference to an external standard calibration curve. Data are reported as means ± standard deviation of triplicate experiments. CML and CEL contents in the samples are expressed as μmol/mmol lysine, μg/g protein, and μg/g sample. 

## 4. Conclusions

The CML and CEL contents in different sections of antler velvet subjected to different processing methods have been simultaneously determined for the first time. The CML contents of antler velvet after freeze-drying, boiling, processing without blood, and processing with blood were 74.55–458.59, 119.44–570.69, 75.36–234.92, and 117.11–456.01 μg/g protein, respectively; the corresponding CEL contents were 0.74–12.66, 11.33–35.93, 0.00–6.75, and 0.00–23.41 μg/g protein. The CML and CEL contents in the same sections of boiled antler velvet were significantly higher than those in freeze-dried antler velvet; high temperatures exacerbate the Maillard reaction, leading to boiled antler velvet producing more CML and CEL. Antler velvet processed with blood had obviously higher CML and CEL contents than antler velvet processed without blood; the antler velvet processed without blood was subjected to physical centrifugal discharge of the blood and therefore contains fewer substrates that can react to generate CML and CEL.

With the same processing methods, the CML and CEL contents were different in different sections of the antler velvet. Wax pieces had significantly higher CML and CEL contents than the types of antler velvet. Closer to the top of the antler velvet, there is a more rapid differentiation capacity and more substrates exist in the antler velvet to produce CML and CEL. Closer to the bottom of the antler velvet, there is a higher degree of ossification and fewer substrates so less CML and CEL is produced.

Through the detection and comparison of CML, CEL, lysine, and protein contents in antler velvet after different processing methods and from various sections, it was found that the different contents of CML and CEL in antler velvet samples are the result of the interaction of protein and lysine at different temperatures.

## Figures and Tables

**Figure 1 molecules-23-03316-f001:**
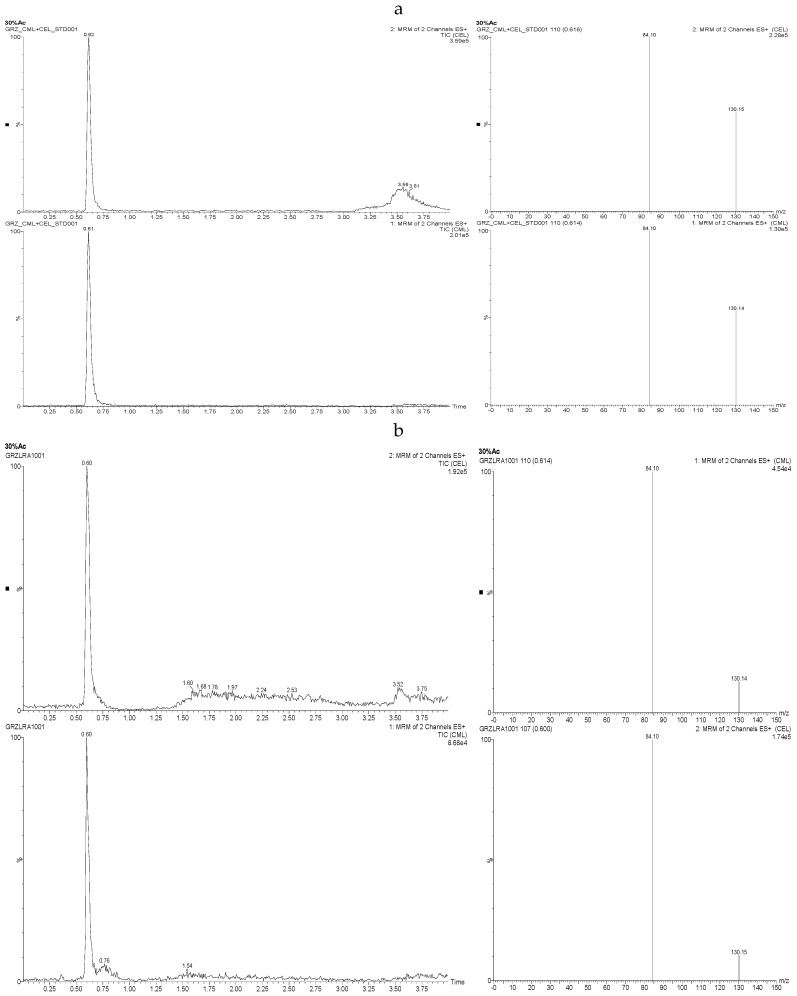
Total ion chromatogram and selected ion of N^ε^-(Carboxymethyl) lysine (CML) and N^ε^-(carboxyethyl) lysine (CEL) standard solutions (**a**) and antler velvet samples (**b**).

**Figure 2 molecules-23-03316-f002:**
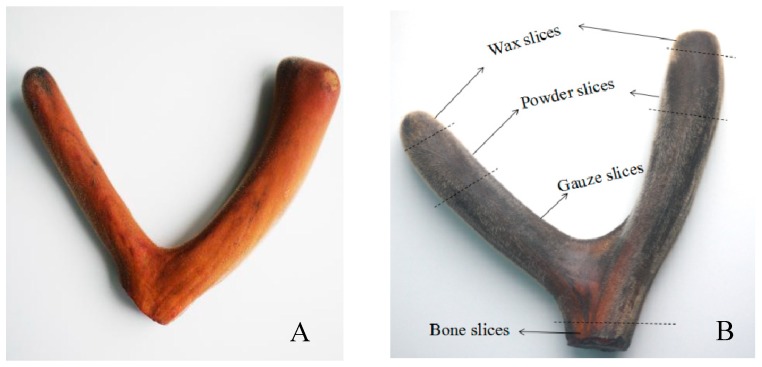
Schematic diagram of fresh antler velvet (**A**) and different sections of processed antler velvet (**B**). The processed antler velvet has a significant Maillard reaction browning compared to the fresh antler velvet. The processed antler velvet was divided into wax slices, powder slices, gauze slices, and bone slices based on morphological and microscopic characteristics.

**Table 1 molecules-23-03316-t001:** Calibration, sensitivity and recovery in UPLC-MS/MS.

Compound	Calibration		Sensitivity		Recovery		
	Range (ng/mL)	R^2^	LOD (ng/g)	LOQ (ng/g)	30 (ng/mL)	300 (ng/mL)	3000 (ng/mL)
CML	20–3500	0.9997	1.3	4.1	95.21 ± 1.22	93.22 ± 1.13	97.42 ± 1.21
CEL	20–3500	0.9987	1.4	4.3	95.43 ± 1.09	93.22 ± 1.24	91.84 ± 1.18

**Table 2 molecules-23-03316-t002:** CML contents in different sections of antler velvet with different processing methods expressed per μg/g protein, μg/g, and μmol/mmol lysine.

Processing Methods	Sections	μg CML/g Protein ^a^	μmol CML/mmol Lysine ^b^	μg CML/g
freeze-dried	wax slices	458.59 ± 22.04	4.00 ± 1.23	328.15 ± 20.13
	powder slices	159.70 ± 11.67	1.71 ± 0.92	94.70 ± 8.72
	gauze slices	97.59 ± 9.22	0.98 ± 0.45	51.47 ± 5.33
	bone slices	74.55 ± 8.94	0.79 ± 0.33	37.51 ± 4.25
	entire	120.93 ± 10.28	1.14 ± 0.72	64.21 ± 6.56
boiled	wax slices	570.69 ± 34.74	6.07 ± 2.82	480.87 ± 31.22
	powder slices	198.64 ± 13.56	2.23 ± 1.42	122.21 ± 9.96
	gauze slices	130.24 ± 10.25	1.38 ± 0.72	71.01 ± 8.23
	bone slices	119.44 ± 10.12	1.18 ± 0.69	51.87 ± 6.09
	entire	141.41 ± 15.23	1.40 ± 0.71	80.09 ± 7.89
processed without blood	wax slices	234.92 ± 23.03	2.58 ± 1.44	200.25 ± 18.27
	powder slices	101.14 ± 12.31	1.59 ± 0.89	89.07 ± 8.93
	gauze slices	99.26 ± 9.18	1.11 ± 0.74	58.46 ± 6.04
	bone slices	75.36 ± 8.56	0.78 ± 0.41	39.13 ± 4.02
	entire	103.14 ± 9.88	1.29 ± 0.82	54.26 ± 5.78
processed with blood	wax slices	456.01 ± 24.32	5.16 ± 2.56	407.88 ± 30.42
	powder slices	167.70 ± 11.82	1.96 ± 0.98	86.79 ± 9.51
	gauze slices	129.02 ± 9.23	1.44 ± 0.74	62.94 ± 7.89
	bone slices	117.11 ± 11.23	1.19 ± 0.70	57.64 ± 6.33
	entire	124.73 ± 12.51	1.30 ± 0.83	73.30 ± 7.88

^a^ Data were calculated using the protein contents quantified by combustion method. ^b^ Data were calculated using the amino acid concentration in the acid hydrolysates, quantified by amino acid analyze.

**Table 3 molecules-23-03316-t003:** CEL contents in different sections of antler velvet with different processing methods, expressed per μg/g protein, μg/g, and μmol/mmol lysine.

Processing Methods	Sections	μg CEL/g Protein ^a^	μmol CEL/mmol Lysine ^b^	μg CEL/g
freeze-dried	wax slices	12.66 ± 1.33	0.11 ± 0.21	9.06 ± 1.12
	powder slices	10.99 ± 0.98	0.10 ± 0.14	6.28 ± 0.74
	gauze slices	1.83 ± 0.32	0.02 ± 0.07	0.96 ± 0.23
	bone slices	0.74 ± 0.12	0.01 ± 0.09	0.36 ± 0.11
	entire	10.84 ± 0.99	0.10 ± 0.11	6.17 ± 0.72
boiled	wax slices	35.93 ± 4.22	0.34 ± 0.23	29.19 ± 4.21
	powder slices	15.43 ± 2.01	0.15 ± 0.11	8.97 ± 1.05
	gauze slices	11.33 ± 1.23	0.13 ± 0.12	6.12 ± 0.72
	bone slices	12.70 ± 1.41	0.11 ± 0.09	6.49 ± 0.78
	entire	14.54 ± 1.47	0.14 ± 0.14	8.87 ± 1.22
processed without blood	wax slices	6.57 ± 0.74	0.06 ± 0.10	5.24 ± 0.56
	powder slices	—	—	—
	gauze slices	—	—	—
	bone slices	—	—	—
	entire	2.57 ± 0.33	0.03 ± 0.06	1.45 ± 0.21
processed with blood	wax slices	23.41 ± 3.01	0.23 ± 0.18	19.22 ± 2.53
	powder slices	2.24 ± 0.22	0.02 ± 0.06	1.37 ± 0.16
	gauze slices	0.03 ± 0.11	0.01 ± 0.04	0.02 ± 0.09
	bone slices	—	—	—
	entire	7.97 ± 0.92	0.08 ± 0.15	4.72 ± 0.52

^a^ Data were calculated using the protein contents quantified by combustion method. ^b^ Data were calculated using the amino acid concentration in the acid hydrolysates, quantified by amino acid analyzer. — Indicates not detected.

**Table 4 molecules-23-03316-t004:** Protein contents in different sections of antler velvet with different processing methods which were determined by combustion method.

Processing Methods	Sections	Protein Contents		Processing Methods	Sections	Protein Contents	
		Content (%)	Coefficient of Variation (%)			Content (%)	Coefficient of Variation (%)
freeze-dried	wax slices	81.56 ± 0.04	0.27	processed without blood	wax slices	79.81 ± 0.09	0.32
	powder slices	57.12 ± 0.03	0.25		powder slices	56.69 ± 0.11	0.28
	gauze slices	52.74 ± 0.10	0.78		gauze slices	58.88 ± 0.31	0.41
	bone slices	49.03 ± 0.25	0.22		bone slices	54.11 ± 0.24	0.33
	entire	56.93 ± 0.34	0.41		entire	56.43 ± 0.28	0.21
boiled	wax slices	81.25 ± 0.12	0.17	processed with blood	wax slices	82.09 ± 0.74	0.56
	powder slices	58.11 ± 0.18	0.14		powder slices	61.49 ± 0.33	0.42
	gauze slices	53.99 ± 0.33	0.21		gauze slices	61.45 ± 0.41	0.38
	bone slices	51.13 ± 0.25	0.25		bone slices	49.65 ± 0.56	0.52
	entire	60.99 ± 0.44	0.33		entire	59.25 ± 0.35	0.41

**Table 5 molecules-23-03316-t005:** Lysine contents in different sections of antler velvet with different processing methods which were determined by amino acid automatic analyzer.

Sample		Lysine Contents g/100 g	Sample		Lysine Contents g/100 g
Processing Methods	Sections		Processing Methods	Sections	
freeze-dried	wax slices	5.87 ± 0.20	processed without blood	wax slices	5.56 ± 0.11
	powder slices	3.96 ± 0.11		powder slices	4.01 ± 0.09
	gauze slices	3.76 ± 0.03		gauze slices	3.76 ± 0.11
	bone slices	3.41 ± 0.09		bone slices	3.58 ± 0.08
	entire	4.02 ± 0.12		entire	3.01 ± 0.15
boiled	wax slices	5.67 ± 0.22	processed with blood	wax slices	5.66 ± 0.12
	powder slices	3.92 ± 0.14		powder slices	4.86 ± 0.07
	gauze slices	3.69 ± 0.10		gauze slices	4.21 ± 0.02
	bone slices	3.15 ± 0.13		bone slices	3.48 ± 0.11
	entire	4.09 ± 0.17		entire	4.03 ± 0.14

**Table 6 molecules-23-03316-t006:** UPLC-MS settings for multiple reaction monitoring (MRM).

Compound	Precursor Ion (*m/z*)	Product Ion (*m/z*)	Cone Voltage (*V*)	Collision Energy (*ev*)	Dwell Time (*ms*)
CML	205	130	25	15	36
	205	84	25	25	36
CEL	219	130	25	15	36
	219	84	25	25	36

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
