# Peer review of "Simultaneous Determination of Nε-(carboxymethyl) Lysine and Nε-(carboxyethyl) Lysine in Different Sections of Antler Velvet after Various Processing Methods by UPLC-MS/MS"

_molecules, 2018, doi:10.3390/molecules23123316_

Reviewer 1 Report

The introduction is short and doesn't fully describe why the authors decided to perform this study. I had to google several things on my own which means the introduction is very thin. I don't understand why blood and bone were mentioned later when they are not described in the introduction.

Please keep in mind that not everyone is familiar with this antler velvet concept thus the authors need to more clearly describe the objectives and provide a figure, perhaps a cartoon of the workflow or something else to orient the readers

The MS experiment seems ok but the discussion and conclusions suffer from the same issues as the introduction. These sections need to be expanded to improve clarity of the material in this paper.

Author Response

Dear editor,

We must thank you and all the reviewers’ valuable comments and thoughtful suggestions. These valuable comments not only helped us with the improvement of our manuscript, but also suggested some neat ideas for our future studies. Please do forward our heartfelt thanks to these experts.

Based on the comments we received, careful modifications have been made to the whole manuscript. All changes made to the text were summarized and were clearly marked in red. In addition, we also consulted a native English speaker for paper revision before submission. We hope the new manuscript will meet your magazine’s standard. Below you will find our point-by-point responses to the editor and the reviewers’ comments. They were summarized in 2 separate documents: 1. Response to reviewer 1’s comments; 2. Other changes.

If you need further information, please do not hesitate to contact me.

Yours sincerely,

Yinshi Sun

Document 1. Responses to reviewer 1’s comments

1. The introduction is short and doesn't fully describe why the authors decided to perform this study. I had to google several things on my own which means the introduction is very thin. I don't understand why blood and bone were mentioned later when they are not described in the introduction.

Response: We are grateful to the reviewer for pointing out our defect, we are sorry that the introduction did not give you enough research background.

In the revised manuscript, we introduced in detail the original intention of our research in the introduction. As a precious Chinese herbal medicine, antler velvet has been used in China, Korea, Japan and Southeast Asia countries for thousands of years and has high medicinal value. However, fresh antler velvet is rich in nutrients, such as proteins and amino acids; these are highly susceptible to spoilage if the antler velvet is not processed promptly. During the processing of antler velvet, advanced glycation end-products (AGEs) were produced by the Maillard reaction. AGEs have been shown to be detrimental to human health as they are closely linked to many diseases. Nε-(carboxymethyl) lysine (CML) and Nε-(carboxyethyl) lysine (CEL) are two typical AGEs and are frequently used as markers of AGE formation, so we decided to determine the content of CML and CEL contents in various sections of antler velvet processed with different methods to estimate the potential consumer intake of AGEs from antler velvet and to provides a foundation and valuable reference for safe antler velvet processing. In addition, we added CML and CEL analysis methods in the introduction(P. 2, L. 58-66).

Besides, we are sorry that the blood and bone mentioned have caused you misunderstanding. In fact, in accordance with the classification of commercially available, antler velvet can be classified as boiled, freeze-dried, processed with blood and without blood by processing methods, and be classified as wax, powder, gauze, and bone slices by sections. Therefore, we choose freeze-dried, boiled, processed without blood and with blood antler velvet, antler velvet after various processing methods were divided into wax, powder, gauze, and bone slices, as the object of this study. The above part has been explained clearly in the introduction.

2. Please keep in mind that not everyone is familiar with this antler velvet concept thus the authors need to more clearly describe the objectives and provide a figure, perhaps a cartoon of the workflow or something else to orient the readers

Response: We are very sorry that we didn’t consider that not everyone is familiar with this antler velvet concept. To orient the readers, We added a figure(Figure 1.) to introduced processed antler velvet has a significant Maillard reaction browning compared to the fresh antler velvet and the processed antler velvet was divided into wax, powder, gauze, and bone slices based on morphological and microscopic characteristics.

3. The MS experiment seems ok but the discussion and conclusions suffer from the same issues as the introduction. These sections need to be expanded to improve clarity of the material in this paper.

Response: We are very happy that the MS experiment has gained your approval. For the short and inadequate issues about the discussion and conclusions, we have expanded to improve it clarity.

The reasons for the differences in CML and CEL contents in different sections of antler velvet after various processing methods are discussed in depth. High temperatures exacerbate the Maillard reaction, leading to the same sections of boiled antler velvet producing more CML and CEL than those in freeze-dried antler velvet. The antler velvet processed without blood was subjected to physical centrifugal discharge of the blood and therefore contains fewer substrates that can react to generate CML and CEL than antler velvet processed with blood.

Closer to the top of the antler velvet, there is a more rapid differentiation capacity and more substrates exist in the antler velvet to produce CML and CEL. Closer to the bottom of the antler velvet, there is a higher degree of ossification and fewer substrates so less CML and CEL is produced. Therefore, from the top to the base of the antler velvet, the CML and CEL contents are gradually reduced.

Factors affecting the CML and CEL contents, including temperatures, content of lysine, proteins, vitamins, and inorganic ions, etc., are discussed. Through the detection and comparison of CML, CEL, lysine, and protein contents in antler velvet after different processing methods and from various sections, it was found that the different contents of CML and CEL in antler velvet samples are the result of the interaction of protein and lysine at different temperatures.

Document 2. Other changes

1. We changed the title to “Simultaneous determination of Nε-(carboxymethyl) lysine and Nε-(carboxyethyl) lysine in different sections of antler velvet after various processing methods by UPLC-MS/MS”. The revised title will reflect the subject of the article clarity.

2. “… by physical means” in section 2.3(P. 2, L. 83-84) was revised as “… by centrifugation” (P. 3, L. 93-94). It was helpful for reader to understand antler velvet processed without blood clarity.

3. “The NIN-derivatized amino acids were detected at 440 nm” in section 2.6(P. 3, L. 104-105) was revised as “The NIN-derivatized lysine were detected at 570 nm” (P. 3, L. 119) . We are very sorry that due to my carelessness, the maximum UV absorption wavelength of NIN-derived lysine is 570nm.

Reviewer 2 Report

Manuscript review: Simultaneous determination of Nε-(carboxymethyl) lysine and Nε-(carboxyethyl) lysine in antler velvet via different processing methods and different parts by UPLC-MS/MS for possible publication in Molecules

Introduction need revision and additional information about analytical methodology should be added. The manuscript type is connected with method development as well as application of novel method for determination of CML and CEL. However it should be noted that the method is well known and already developed for other sample type but the protocol is the same  at the step of determination of CML and CEL. It is necessary here to add detailed information about actual methodology and information what is new in this research. If it is not applied method LC/MSMS it should be sample type eg. here Food Chemistry 269, 15, Pages 466-472.

Section 2.7 is this new procedure or modified procedure already developed ?

Section 2.8 please add detailed information about columns (necessary to compare it in 3.2 section).

- the MRM mode was used only ? Did Authors have an experience with SIM mode ?

- l.128-129 multiple information about MRM

- it is better to use ug/g or umol/mmol

section 3.2 l.152 please prove it through chromatographic parameters description

table 1 - please add SD to recovery values

Section discussion should be added and developed to the manuscript

Conclusion need addition and revision.

Author Response

Dear editor,

We must thank you and all the reviewers’ valuable comments and thoughtful suggestions. These valuable comments not only helped us with the improvement of our manuscript, but also suggested some neat ideas for our future studies. Please do forward our heartfelt thanks to these experts.

Based on the comments we received, careful modifications have been made to the whole manuscript. All changes made to the text were summarized and were clearly marked in red. In addition, we also consulted a native English speaker for paper revision before submission. We hope the new manuscript will meet your magazine’s standard. Below you will find our point-by-point responses to the editor and the reviewers’ comments. They were summarized in 2 separate documents: 1. Response to reviewer 2’s comments; 2. Other changes.

If you need further information, please do not hesitate to contact me.

Yours sincerely,

Yinshi Sun

Document 1. Responses to reviewer 2’s comments

1. Introduction need revision and additional information about analytical methodology should be added. The manuscript type is connected with method development as well as application of novel method for determination of CML and CEL. However it should be noted that the method is well known and already developed for other sample type but the protocol is the same at the step of determination of CML and CEL. It is necessary here to add detailed information about actual methodology and information what is new in this research. If it is not applied method LC/MSMS it should be sample type eg. here Food Chemistry 269, 15, Pages 466-472.

Response: We are grateful to the reviewer for pointing out our defect, we are sorry that the introduction did not give you enough research background. In the revised manuscript, we introduced in detail the original intention of our research in the introduction, and we added CML and CEL analysis methods in the introduction(P. 2, L. 58-66).

Although methods for simultaneous determination of CML and CEL have been developed for the determination of other sample types, such as milk, chocolate and barbecue. However, details on the contents of CML and CEL of antler velvet are lacking. As a precious Chinese herbal medicine, antler velvet has been used in China, Korea, Japan and Southeast Asia countries for thousands of years and has high medicinal value. Antler velvet is divided into boiled, freeze-dried, processed with blood and without blood based on processing methods, and is divided into wax, powder, gauze, and bone slices by sections. We reported the content of CML and CEL in different sections of antler velvet after various processing methods for the first time. These study can estimates the potential consumer intake of AGEs from antler velvet and can also provides a foundation and valuable reference for safe antler velvet processing.

Besides, we used WATERS CORTECS HILIC UPLC column to separate CML and CEL from antler velvet samples for the first time. HILIC column uses the separation principle of affinity chromatography to maximize the retention and separation of highly polar compounds, such as CML and CEL, relative to C18 and HSST3 columns. This part has been discussed in section 3.2(P. 5, L. 166-172).

2. Section 2.7 is this new procedure or modified procedure already developed ?

Response: We are grateful to the reviewer for pointing out our defect. Section 2.7 is a modified procedure already developed, we refer to the predecessors' processing methods. And the reference has been added to the corresponding positions in the revised manuscript(P. 3, L. 124).

3. Section 2.8 please add detailed information about columns (necessary to compare it in 3.2 section).

- the MRM mode was used only ? Did Authors have an experience with SIM mode ?

- l.128-129 multiple information about MRM

- it is better to use ug/g or umol/mmol

Response: We are grateful to the reviewer for pointing out our oversight. Detailed information of column has been added to section 2.8(P. 4, L. 140-141) , and we compared the advantages of HILIC compared to other columns in section 3.2(P. 5, L. 166-172).

- The MRM mode was only used to simultaneous determination of CML and CEL in antler velvet samples. We have experiences with SIM and MRM, SIM for primary MS, only one ion is monitored; MRM for secondary or multi-stage MS, precursor ion and multiple product ions are monitored. Since CML and CEL are homologues, MRM has the advantages of anti-interference and high sensitivity, which is suitable for simultaneous determination of CML and CEL. And the MRM model is used in the reference to monitor CML and CEL.

- Multiple information about MRM has been added in table 1(P. 4, L. 152).

- We accept the reviewer's opinion and use μmol/mmol lysine, μg/g protein, and μg/g sample to express CML and CEL contents in antler velvet samples.

1. section 3.2 l.152 please prove it through chromatographic parameters description

Response: We are grateful to the reviewer for pointing out our defect. We have proved that UPLC-MS chromatograms using water and acetonitrile as the eluent have better spectrum peak symmetries and fewer miscellaneous peaks than water and methanol through chromatographic parameters description (P. 5, L. 172-176).

2. table 1 - please add SD to recovery values

Response: We are grateful to the reviewer for pointing out our defect. We have added SD to recovery values in table 1(P. 6, L. 196).

3. Section discussion should be added and developed to the manuscript

Response: We are grateful to the reviewer for pointing out our shortage. We have added and developed results and discussion in revised manuscript to improve it clarity.

In section 3.4.1(P. 7, L. 227-231), we discussed the reasons for the differences in CML content in different sections of processed antler velvet. The differentiation degree of antler velvet cells in different sections is different, and the difference in substrate concentration, such as reducing sugar, amino acids and protein involved in Maillard reaction is the main reason for the differences of CML content in different sections of processed antler velvet.

In section 3.4.2(P. 8, L. 250-256), we described in detail the differences in CEL content in different sections of antler velvet after various processing methods, and explain the reasons for this differences. The differences in temperature and contents of proteins, amino acids, and reducing sugars are the main reasons for the different CEL contents after different processing methods.

In section 3.6(P. 10, L. 294-301), we discussed factors affecting CML and CEL contents in different sections of antler velvet after various processing methods, including temperature, content of lysine and protein. High temperatures exacerbate the Maillard reaction and cause boiled antler velvet to produce more CML and CEL than freeze-dried antler velvet. Lysine and proteins are substrates for the Maillard reaction, antler velvet processed with blood, which is rich in lysine and protein, produce more CML and CEL than antler velvet processed without blood. Similarly, wax pieces are more likely to produce CML and CEL than other sections.

4. Conclusion need addition and revision.

Response: We are grateful to the reviewer for pointing out our shortage. we have expanded and revised conclusion to improve it clarity.

The reasons for the differences in CML and CEL contents in different sections of antler velvet after various processing methods are discussed in depth. High temperatures exacerbate the Maillard reaction, leading to the same sections of boiled antler velvet producing more CML and CEL than those in freeze-dried antler velvet. The antler velvet processed without blood was subjected to physical centrifugal discharge of the blood and therefore contains fewer substrates that can react to generate CML and CEL than antler velvet processed with blood.

Closer to the top of the antler velvet, there is a more rapid differentiation capacity and more substrates exist in the antler velvet to produce CML and CEL. Closer to the bottom of the antler velvet, there is a higher degree of ossification and fewer substrates so less CML and CEL is produced. Therefore, from the top to the base of the antler velvet, the CML and CEL contents are gradually reduced.

Factors affecting the CML and CEL contents, including temperatures, content of lysine, proteins, vitamins, and inorganic ions, etc., are discussed. Through the detection and comparison of CML, CEL, lysine, and protein contents in antler velvet after different processing methods and from various sections, it was found that the different contents of CML and CEL in antler velvet samples are the result of the interaction of protein and lysine at different temperatures.

Document 2. Other changes

1. We changed the title to “Simultaneous determination of Nε-(carboxymethyl) lysine and Nε-(carboxyethyl) lysine in different sections of antler velvet after various processing methods by UPLC-MS/MS”. The revised title will reflect the subject of the article clarity.

2. “… by physical means” in section 2.3(P. 2, L. 83-84) was revised as “… by centrifugation” (P. 3, L. 93-94). It was helpful for reader to understand antler velvet processed without blood clarity.

3. “The NIN-derivatized amino acids were detected at 440 nm” in section 2.6(P. 3, L. 104-105) was revised as “The NIN-derivatized lysine were detected at 570 nm” (P. 3, L. 119) . We are very sorry that due to my carelessness, the maximum UV absorption wavelength of NIN-derived lysine is 570nm.

Round  2

Reviewer 1 Report

There is now more background info in the introduction and the discussion is clearer. Figure 1 helps with orienting the readers and the language has improved in the paper.

Reviewer 2 Report

Authors revised and correct the manuscript correctly, ms is very interisting indeed.